# Domain-swapped T cell receptors improve the safety of TCR gene therapy

Michael T Bethune[1], Marvin H Gee[2], Mario Bunse[3], Mark S Lee[4,5], Eric H Gschweng[6], Meghana S Pagadala[1], Jing Zhou[7], Donghui Cheng[8], James R Heath[7], Donald B Kohn[6], Michael S Kuhns[4,5], Wolfgang Uckert[3,9], David Baltimore[1]*

[1]Division of Biology and Biological Engineering, California Institute of Technology, Pasadena, United States; [2]Program in Immunology, Stanford University School of Medicine, Stanford, United States; [3]Max Delbrück Center for Molecular Medicine, Berlin, Germany; [4]Department of Immunobiology, University of Arizona, Tucson, United States; [5]The BIO5 Institute, University of Arizona, Tucson, United States; [6]Department of Microbiology, Immunology, and Molecular Genetics, University of California, Los Angeles, Los Angeles, United States; [7]Division of Chemistry and Chemical Engineering, California Institute of Technology, Pasadena, United States; [8]Eli and Edythe Broad Center of Regenerative Medicine and Stem Cell Research, University of California, Los Angeles, Los Angeles, United States; [9]Institute of Biology, Humboldt-Universität zu Berlin, Berlin, Germany

*For correspondence: baltimo@ caltech.edu

**Abstract** T cells engineered to express a tumor-specific αβ T cell receptor (TCR) mediate anti-tumor immunity. However, mispairing of the therapeutic αβ chains with endogenous αβ chains reduces therapeutic TCR surface expression and generates self-reactive TCRs. We report a general strategy to prevent TCR mispairing: swapping constant domains between the α and β chains of a therapeutic TCR. When paired, domain-swapped (ds)TCRs assemble with CD3, express on the cell surface, and mediate antigen-specific T cell responses. By contrast, dsTCR chains mispaired with endogenous chains cannot properly assemble with CD3 or signal, preventing autoimmunity. We validate this approach in cell-based assays and in a mouse model of TCR gene transfer-induced graft-versus-host disease. We also validate a related approach whereby replacement of αβ TCR domains with corresponding γδ TCR domains yields a functional TCR that does not mispair. This work enables the design of safer TCR gene therapies for cancer immunotherapy.

## Introduction

T lymphocytes recognize cellular targets with extraordinary sensitivity and specificity. The T cell receptor (TCR) on the T cell surface binds to its cognate peptide antigen presented on major histocompatibility complex (MHC) molecules on the target cell surface (*Reinherz, 2015*). The TCR - a heterodimer that is uniquely rearranged in each T cell during its development - is the sole determinant of T cell specificity (*Dembić et al., 1986*). Accordingly, transfer of the α and β genes encoding a tumor-reactive αβ TCR can impart anti-tumor reactivity to a cancer patient's T cells (*Clay et al., 1999*). Upon reinfusion into the patient, these engineered T cells mediate potent anti-tumor cytotoxicity (*Park et al., 2011*). This approach – termed TCR gene therapy – has demonstrated clinical responses for several malignancies (*Johnson et al., 2009*; *Morgan et al., 2006*; *Davis, 2010*; *Parkhurst, 2011*; *Robbins et al., 2015*, *2011*), which may be bolstered in the future through combination with complementary immunotherapies like dendritic cell-based vaccination and checkpoint

**eLife digest** T cells enable the immune system to recognize invading microbes and diseased cells while ignoring healthy cells. The ability of a T cell to recognize a specific microbe or diseased cell is determined by two proteins that pair to form its "T cell receptor." The paired receptors are exported to the surface of the T cell, where they bind to infected or cancerous cells. Those T cells that produce receptors that bind healthy cells are eliminated during development.

T cells can generally distinguish between the body's own cells and the cells of invading bacteria or other microbes. However, cancer cells are more difficult to identify because they are similar to healthy cells. Efforts to develop therapies that enhance the immune system's ability to recognize cancer cells have had only limited success. One successful approach – known as T cell receptor gene therapy – modifies T cells to destroy cancer cells by arming them with a cancer-specific T cell receptor.

This technique produces T cells possessing two T cell receptors – the cancer-specific receptor and the one it had originally – so it is possible for proteins from the two receptors to mispair. This impedes the correct pairing of the cancer-specific T cell receptor, reducing the effectiveness of the therapy. More importantly, mispaired T cell receptors may cause the immune cells to attack healthy cells in the body, leading to autoimmune disease. To make T cell receptor gene therapy safe, the cancer-specific receptor must not mispair with the resident receptor.

Here, Bethune et al. describe a new strategy to prevent T cell receptors from mispairing. The researchers altered the arrangement of particular regions in a cancer-specific T cell receptor to make a new receptor called a domain-swapped T cell receptor (dsTCR). Like normal T cell receptors, the dsTCRs were exported to the T cell surface and were able to interact with other proteins involved in immune responses. Furthermore, T cells armed with dsTCRs were able to kill cancer cells and prevent tumor growth in mice. Unlike other cancer-specific receptors, dsTCRs did not mispair with the resident T cell receptors in mouse or human cells, and did not cause autoimmune disease in mice.

The findings of Bethune et al. show that the structure of the T cell receptor is unexpectedly robust, in that it still works even if it is modified. The next step is to study dsTCRs in more detail with the aim of optimizing them so that they might be used in human clinical trials in the future.

blockade (*Abate-Daga et al., 2013*; *Brahmer and Hammers, 2015*; *Mellman et al., 2011*; *Sharma and Allison, 2015*; *Topalian et al., 2015*; *Chodon, 2014*).

Despite the promise of TCR gene therapy, there remain significant limitations to its safety and efficacy. Among these is that TCR-transduced $\alpha\beta$ T cells express two $\alpha$ and two $\beta$ chains, a non-physiological situation in which the introduced TCR $\alpha$ and $\beta$ chains can mispair with the endogenous TCR $\beta$ and $\alpha$ chains, respectively. TCR chain mispairing reduces the level of correctly-paired, tumor-reactive TCR heterodimers expressed on the T cell surface, possibly reducing therapeutic efficacy (*Jorritsma et al., 2007*). More ominously, mispairing can generate self-reactive TCR heterodimers that do not undergo thymic negative selection. These mispaired TCRs engender autoreactivity in TCR-transduced human T cells *in vitro* (*van Loenen et al., 2010*) and mediate lethal graft-versus-host disease in mice administered TCR-transduced T cells following a protocol mimicking human clinical trials (*Bendle et al., 2010*; *Bunse, 2014*). Although no serious adverse events have been attributed to TCR mispairing in engineered T cell trials (*Rosenberg, 2010*), autoreactive off-target and off-tumor engineered T cell responses have caused deaths (*Linette et al., 2013*; *Morgan et al., 2013*, *2010*). These underscore the need to safeguard against TCR mispairing-related autoreactivity, particularly as more potent immunotherapy regimes are employed.

Efforts to prevent TCR mispairing can be broadly categorized as either engineering the transduced TCR (adding interchain disulfide bonds, murinizing portions of the TCR, expressing TCR as a single chain) (*Uckert and Schumacher, 2009*) or reducing expression of the endogenous TCR (shRNA knockdown (*Bunse, 2014*; *Okamoto et al., 2009*) or genomic knockout [*Provasi et al., 2012*]). Although several engineering strategies improve pairing between the transduced chains, complete prevention of mispairing has not been achieved (*Thomas et al., 2007*) and murine TCRs

are immunogenic (*Davis, 2010*). Endogenous TCR knockout prevents mispairing, but the extensive processing currently required to generate these cells is incompatible with clinical protocols. The ideal solution will prevent mispairing entirely, eliminating the risk of autoimmunity. Additionally, modifications made to the introduced TCR chains should avoid foreign sequences to minimize immunogenicity. Finally, these modifications must be restricted to the constant TCR domains, such that they can be applied without further optimization to any TCR of therapeutic interest. We describe a novel approach for preventing TCR mispairing that meets these criteria. We show that this approach is further improved by combining it with the complementary strategy of endogenous TCR knockdown.

## Results

### Domain-swapped TCR (dsTCR) design

Our approach to prevent TCR mispairing exploits the molecular requirements for TCR biogenesis and function. The TCR α and β chains each contain a membrane-distal variable immunoglobulin domain (V), which imparts specificity, and several constant domains including a membrane-proximal constant immunoglobulin domain (C), a connecting peptide (cp), a transmembrane helix (TM), and a short cytoplasmic tail (cyto) (*Figure 1A*). To achieve functional form, the TCR αβ heterodimer must assemble with six additional chains (CD3 dimers γε, δε, and ζ2), which facilitate export of the TCR

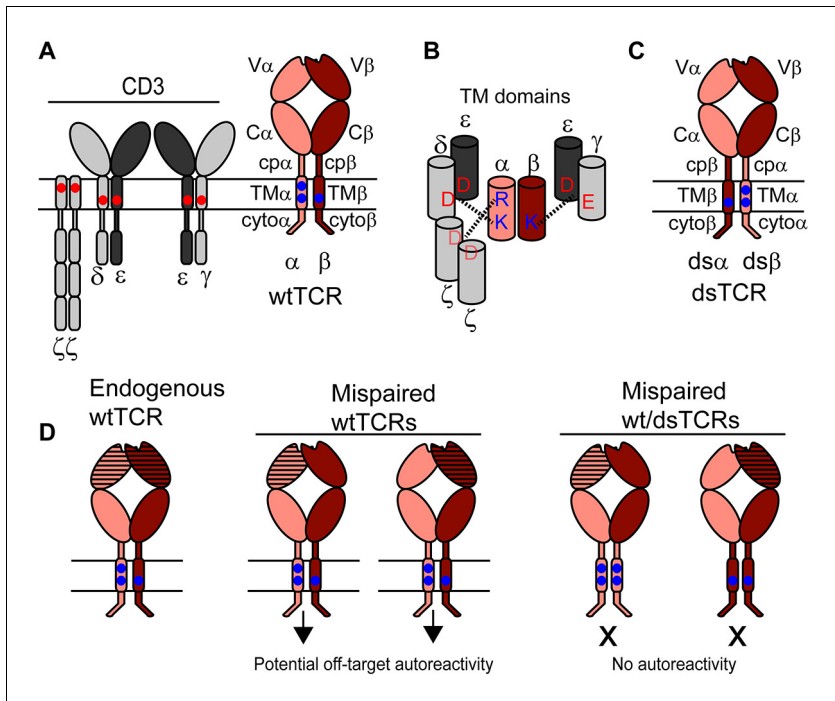

**Figure 1.** Schematic outlining the domain-swapped TCR strategy. (**A**) The TCR/CD3 complex comprises the antigen-specific variable (**V**) Ig domain and constant domains (constant Ig, C; connecting peptide, cp; transmembrane helix, TM; and cytoplasmic tail, cyto), which assemble with CD3 chains. CD3 chains are required for export of the TCR/CD3 complex to the cell surface and for signaling. Parallel horizontal lines represent the cell membrane. (**B**) Schematic showing key interactions between basic residues in the TCR TM domain and acidic residues in the CD3 TM domains. (**C**) Domain-swapped TCRs retain all domains of the wild-type TCR with altered covalent connectivity. (**D**) Mispairing between therapeutic and endogenous TCR chains can result in autoreactivity (middle). Mispairing between domain-swapped therapeutic and endogenous TCR chains will result in heterodimers that lack constant domains needed to assemble with CD3, preventing surface expression and function of potentially autoreactive TCRs (right). Domain-swapped TCRs are thus expected to be functional but incapable of mediating mispairing-related autoreactivity.

complex to the cell surface and mediate signal transduction upon antigen binding (*Call and Wucherpfennig, 2005*). If the TCR/CD3 complex is not assembled properly prior to export, it is degraded (*Bonifacino, 1989*). Assembly with CD3 requires contacts within the constant domains of both the TCR α and β chains (*Call et al., 2002*; *Kuhns and Davis, 2007*; *Xu and Call, 2006*), most critically the basic residues within the transmembrane domains (*Call et al., 2002*)(*Figure 1B*). We designed interchain domain-swapped (ds) TCRs in which select constant domains of the TCR α and β chains are exchanged in a reciprocal manner (*Figure 1C*). Correctly paired α/β dsTCRs retain all domains necessary to assemble with CD3 and to enact tumor-targeted immunity. By contrast, mispaired heterodimers comprising one dsTCR chain and one wild-type (wt) TCR chain lack domains necessary to assemble with CD3 or to enact autoimmune responses (*Figure 1d*).

## Domain-swapped TCRs (dsTCR) assemble with CD3 and express on the cell surface

We designed three dsTCR architectures in which α and β constant domains were swapped at the V-C junctions (dsTCR$_V$), at the C-cp junctions (dsTCR$_C$), or at the cp-TM junctions (dsTCR$_{cp}$) (*Figure 2A*). Initially, we generated these dsTCR variants for the HLA-A2-restricted, MART1-specific F5 TCR (*Johnson et al., 2006*) (*Figure 2—figure supplement 1*). We transfected CD3$^+$ 293T cells

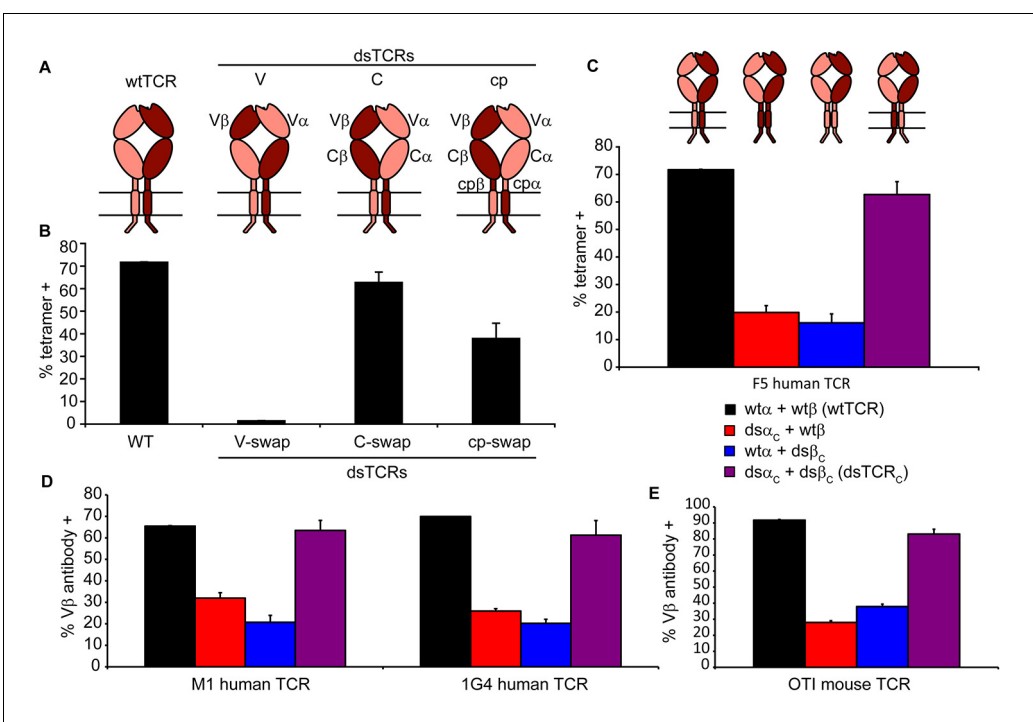

**Figure 2.** Domain-swapped TCRs assemble with CD3 and retain antigenic specificity. (**A**) Schematic of dsTCR architectures. Domains are swapped following the V domains (dsTCR$_V$), C domains (dsTCR$_C$), or connecting peptides (dsTCR$_{cp}$). (**B**) Flow cytometric analysis of peptide-MHC multimer binding by CD3+ 293T transfected with F5 wtTCR or dsTCRs. (**C**) Flow cytometric analysis comparing peptide-MHC multimer binding by CD3+ 293T transfected with F5 wtTCR, dsTCR$_C$, or simulated mispaired constructs. Schematics of constructs are above corresponding data. (**D–E**) Flow cytometric analysis comparing anti-V$_β$ antibody binding by CD3 + 293T transfected with wtTCR, dsTCR$_C$, or simulated mispaired constructs for (**D**) two additional human TCRs and (**E**) mouse OTI TCR. Means ± SD for two replicate transfections are shown.

The following figure supplements are available for figure 2:

**Figure supplement 1.** Amino acid sequences of initial domain-swapped TCR construct chimeric junctions.

**Figure supplement 2.** Optimization of domain-swapped junctions for dsTCR$_V$ and dsTCR$_{cp}$.

with each F5 TCR variant and evaluated A2/MART1 tetramer binding by flow cytometry. The dsTCR$_V$ construct did not bind tetramer (*Figure 2B*), and variation of the precise position at which the V domains were exchanged did not rectify this (*Figure 2—figure supplement 2A*). However, both F5 dsTCR$_C$ and dsTCR$_{cp}$ bound tetramer, indicating these architectures express on the cell surface and retain specificity for their cognate ligand (*Figure 2B*). Cells transfected with F5 dsTCR$_C$ bound tetramer with comparable avidity to wtTCR, whereas tetramer binding for dsTCR$_{cp}$ was lower. We varied the position at which the domains were exchanged in the dsTCR$_{cp}$ architecture to identify the optimal design for this construct (α1β5, *Figure 2—figure supplement 2B*). The dsTCR$_C$ and optimized dsTCR$_{cp}$ constructs were selected for further investigation.

We posit that dsTCRs are safer than wtTCR because mispaired TCRs comprising one ds and one wtTCR chain should have impaired capacity to assemble with CD3 and express on the cell surface. To determine whether dsTCR chains properly assemble CD3 when mispaired with a wtTCR chain, we prepared hybrid constructs in which only the α or β chain of F5 TCR was domain-swapped at the C-cp junction, and tested for expression on the surface of CD3$^+$ 293T (*Figure 2C*). As expected, compared to fully wtTCR or dsTCR$_C$, surface expression of these simulated mispaired constructs was significantly reduced, indicating impaired assembly of CD3.

To have broad therapeutic potential, our approach must be applicable to any human TCR of clinical interest without further optimization. Ideally, this approach should also be applicable to murine TCRs, enabling preclinical studies of safety and efficacy. We constructed dsTCR$_C$ and hybrid wt/ds variants of two additional clinically relevant human TCRs – HLA-A2/MART1-specific M1 TCR (*Chhabra et al., 2008*) and HLA-A2/NY-ESO-1-specific 1 G4 TCR (*Chen et al., 2000*)–as well as the H-2Kb/ovalbumin-specific OTI murine TCR. For both additional human TCRs as well as the murine TCR, dsTCR derivatives express on the surface of CD3$^+$ 293T and mitigate mispairing with wtTCR chains (*Figure 2D,E*), suggesting this approach can be applied generally to human and mouse TCRs.

## dsTCRs are functional in lymphocytes

Altering the spatial relationship between constant TCR domains could potentially alter the orientation of CD3 chains in the TCR/CD3 complex, impairing its function. To compare the orientation of CD3 chains around wt- and dsTCRs, we used a dimerization reporter assay previously employed to determine the orientation of CD3 chains around the αβ wtTCR (*Kuhns et al., 2010*). The murine pro-B cell line, BaF3, will grow in the absence of IL-3 only when provided Jak-Stat signaling via dimerized signaling domains of the erythropoietin receptor (EpoR). A fusion of CD3ε to EpoR (εE) was previously shown to drive proliferation of BaF3 when incorporated in the TCR/CD3 complex, indicating that the CD3 δε and γε dimers cluster on one face of the wtTCR, juxtaposing the CD3ε chains (*Kuhns et al., 2010*) (*Figure 3A*). We transduced BaF3 cells with all CD3 chains, including εE, and either human F5 wtTCR, dsTCR$_C$, or dsTCR$_{cp}$ and then measured TCR/CD3 complex on the cell surface. All three TCRs formed complexes with CD3/εE and bound cognate peptide-MHC multimer (*Figure 3—figure supplement 1*). Following removal of IL-3 from the cell media, εE drove proliferation of BaF3 cells when complexed with wtTCR, dsTCR$_C$, or dsTCR$_{cp}$, indicating that dsTCRs assemble with CD3 chains in the same orientation as wtTCR (*Figure 3B*). Results were similar for the murine OTI TCR (*Figure 3C* and *Figure 3—figure supplement 1*).

In both the 293T and BaF3 assays, CD3 chains are expressed at high, non-physiological levels. We have observed that some TCR is exported to the surface of transfected 293Ts even when CD3 assembly is incomplete (i.e. 26%, 0%, 52%, and 100% residual export respectively when omitting CD3δ, CD3ε, CD3γ, or CD3ζ from transfection). We suspected that this accounted for the low level of apparent pairing between dsTCR and wtTCR chains observed on 293T cells (*Figure 2C–E*). To determine whether dsTCRs form a complex with endogenous CD3 in T cells, we transduced the Jurkat human T cell line with wt- and dsTCRs and used flow cytometry to assess surface expression. Both dsTCR$_C$ and dsTCR$_{cp}$ assemble endogenous CD3, express on the cell surface, and bind cognate A2/MART1 multimers to an extent similar to the wtTCR (*Figure 3D*). By contrast, hybrid constructs comprising one dsTCR chain and one wtTCR chain were not detectable on the surface of transduced Jurkat cells, indicating that, as expected, these mispaired constructs failed to form a mature complex (*Figure 3E*). Virtually all transduced (Myc271$^+$) cells were positive for A2/MART1 staining (*Figure 3F*), indicating the dsTCRs competed favorably for CD3 recruitment in the presence of the endogenous Jurkat TCR. To determine whether dsTCR/CD3 complexes are functional, dsTCR-transduced Jurkat T cells were coincubated with K562 target cells expressing either cognate A2/

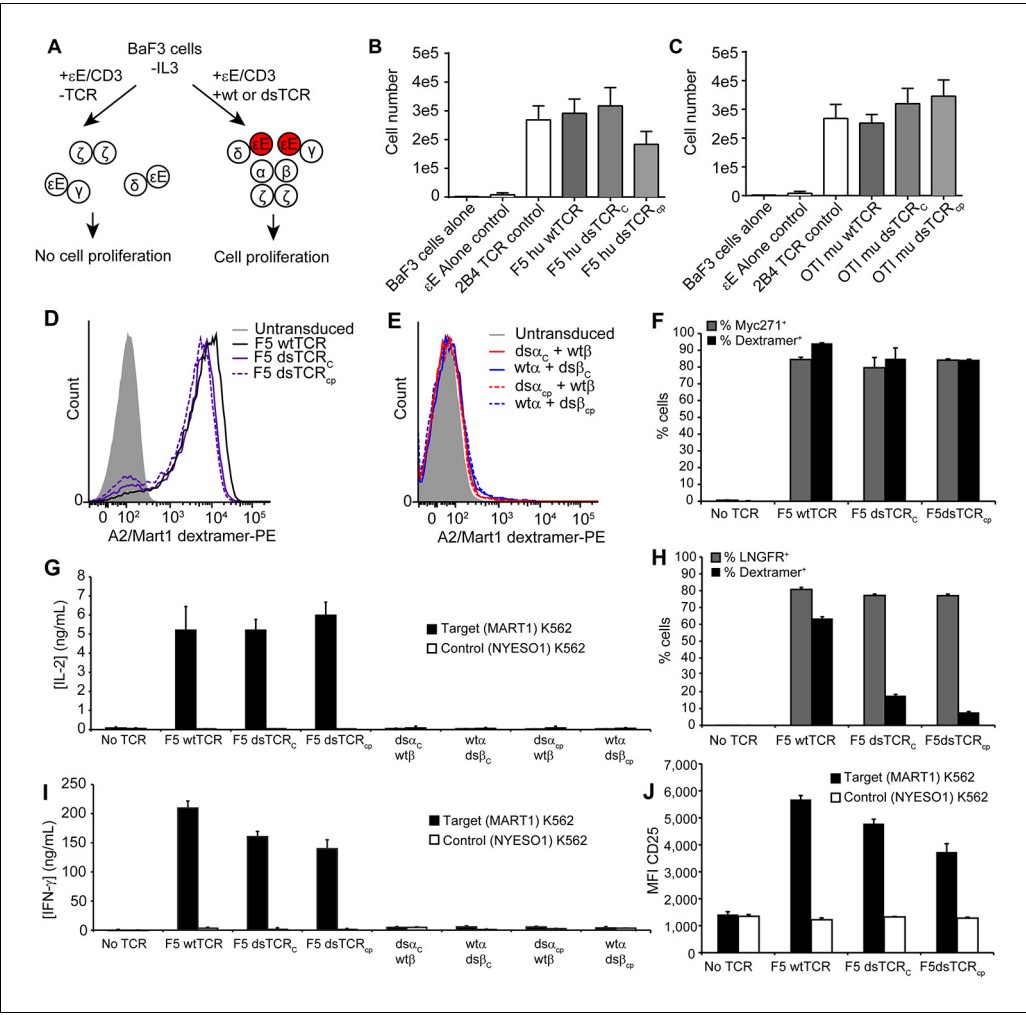

**Figure 3.** Domain-swapped TCRs are functional in lymphocytes. (**A**) Schematic of the BaF3 proliferation assay for determining CD3 orientation within the TCR/CD3 complex. CD3ε-EpoR fusion proteins promote IL-3-independent proliferation of BaF3 only if assembly in the TCR/CD3 complex orients them in juxtaposition to one another. (**B-C**) CD3ε-EpoR-driven proliferation of BaF3 cells expressing wild-type and domain-swapped (**B**) human F5 TCR and (**C**) mouse OTI TCR. Cell number was measured 72 hr after IL-3 deprivation. Means ± SD for 3 independent experiments are shown. (**D-E**) Representative flow cytometry histograms comparing peptide-MHC multimer binding by Jurkat T cells transduced with (**D**) F5 wtTCR or dsTCR constructs or (**E**) simulated mispaired constructs. Multimer binding is reliant on cell surface expression following assembly with endogenous CD3 chains. (**F**) Flow cytometric measurement of TCR and Myc271 expressed on the surface of transduced Jurkats. Myc271 is an independent transduction marker expressed from the same vector as TCR. (**G**) ELISA measuring secretion of IL-2 from TCR-transduced Jurkats following 48 hr coincubation with cognate or control antigen-expressing K562 target cells. (**H**) Flow cytometric measurement of dextramer-binding TCR and LNGFR expressed on the surface of transduced primary human T cells. LNGFR is an independent transduction marker expressed from the same vector as TCR. (**I**) ELISA measurement of secreted IFN-γ and (**J**) flow cytometric measurement of CD25 expressed on the T cell surface from TCR-transduced primary human T cells following 48 hr coincubation with cognate or control antigen-expressing K562 target cells. For panels (**f-j**), means ± SD for 3 technical replicates are shown.

The following figure supplement is available for figure 3:

**Figure supplement 1.** Domain-swapped TCRs are expressed on the surface of CD3[+] BaF3 cells.

MART1 or control A2/NYESO1 single-chain trimer (*Hansen et al., 2009*). Consistent with multimer staining results, dsTCRs mediated antigen-specific release of IL-2, whereas hybrid wt/dsTCRs simulating mispairing did not (*Figure 3G*).

Unlike the Jurkat T cell line, primary T cells express a diverse repertoire of endogenous T cell receptors, which may mispair or compete for surface expression with the transduced TCR to differing extents across cells. We therefore transduced primary T cells with wt- and dsTCRs to assess their function in a more physiologically relevant context. As expected, wtTCR was detected in only a subset (74%) of transduced (LNGFR$^+$) cells, indicating ~25% of T cells express an endogenous TCR that out-competes F5 TCR for CD3 recruitment and cell surface expression (*Figure 3H*). Dextramer staining was lower for dsTCRs than wtTCRs in primary T cells (25% and 11% of LNGFR+ cells, for dsTCR$_C$ and dsTCR$_{cp}$, respectively; MFI = 21% and 13% that of wtTCR). This indicates a larger subset of endogenous TCRs outcompete the dsTCRs for CD3 recruitment. Nonetheless, T cells transduced with either wtTCR or dsTCRs were functional, secreting IFN-$\gamma$ and upregulating cell surface CD25 in an antigen-specific manner when coincubated with target cells (*Figure 3I,J*). As with Jurkat cells, simulated mispaired constructs elicited no such activation (*Figure 3I,J*). Thus, dsTCRs assemble with endogenous CD3 to mediate antigen-specific immunity in primary lymphocytes and are rendered nonfunctional when mispaired with wtTCR chains.

## dsTCRs do not mispair with endogenous TCR and do not cause graft-versus-host disease

Transduced $\alpha$ and $\beta$ chains can each mispair with endogenous chains, and autoreactivity is an unpredictable result of the particular transduced and endogenous chains involved (*van Loenen et al., 2010*). To investigate whether dsTCR chains can mispair with any of the diverse array of endogenous TCR chains expressed across primary T cells, we devised an assay for measuring mispairing without regard to antigenic specificity. We tagged the N-termini of the $\alpha$ and $\beta$ chains of F5 TCR with cMyc and V5 epitope tags, respectively, and confirmed that the addition of these tags did not alter TCR surface expression on Jurkat T cells (*Figure 4—figure supplement 1*). We then delivered each chain individually (i.e. Myc-$\alpha$ only or V5-$\beta$ only), such that the export of the tagged TCR chain is dependent on mispairing with endogenous TCR chains. The wt$\alpha$ and wt$\beta$ chains of F5 TCR mispaired with endogenous TCR chains in 88% and 58% of transduced T cells, respectively. By contrast, neither ds$\alpha$ nor ds$\beta$ chains exhibited detectable mispairing with endogenous TCR chains (*Figure 4A*). Similar results were obtained with the murine OTI TCR: wt$\alpha$ and wt$\beta$ chains were mispaired in 63% and 84% of transduced murine T cells, whereas mispairing was not observed with ds$\alpha$ or ds$\beta$ chains (*Figure 4B*). Thus, even when mispairing propensity is maximized by transduction of a single dsTCR chain in the absence of stoichiometric expression of its correct partner, dsTCR chains exhibit no detectable mispairing with endogenous TCR chains.

To determine whether dsTCRs prevent deleterious autoimmunity resulting from TCR mispairing, we used a previously reported mouse model for TCR gene transfer-induced graft-versus-host disease (TI-GvHD) (*Bendle et al., 2010*). Mice that develop TI-GvHD due to TCR chain mispairing exhibit destruction of the hematopoietic compartment and rapid weight loss, generally necessitating sacrifice within 1–2 days of onset of cachexia. We transduced T cells with wt- and dsTCR derivatives of OTI TCR, as this TCR resulted in the highest incidence of TI-GvHD of those TCRs tested in the initial report of the model. Based on our finding that dsTCR chains do not mispair with endogenous TCR chains, we expected to recapitulate TI-GvHD in wtTCR-transduced T cell recipients, but not in dsTCR-transduced T cell recipients. Indeed, mice that received wtTCR(IRES)-transduced T cells - in which the transduced wtTCR chains were linked by an internal ribosome entry site (IRES) sequence – exhibited significant cachexia following IL-2 administration (*Figure 4C*). Although some of these mice regained weight and survived, 47% (7/15) developed TI-GvHD to an extent that necessitated euthanasia (*Figure 4D*). Consistent with previous results (*Bendle et al., 2010*), the incidence of TI-GvHD could be reduced but not eliminated by replacing the IRES with a viral 2A sequence to ensure stoichiometric expression of the wtTCR chains. In this wtTCR(2A) group, less weight loss was observed and only 1/15 mice developed fatal TI-GvHD. By contrast, cachexia was not observed for mice that received dsTCR$_C$(2A)-transduced T cells, and none (0/25) of these mice developed TI-GvHD. Thus, dsTCRs prevent mispairing and thereby prevent mispairing-related autoimmune complications in a model of TCR gene therapy.

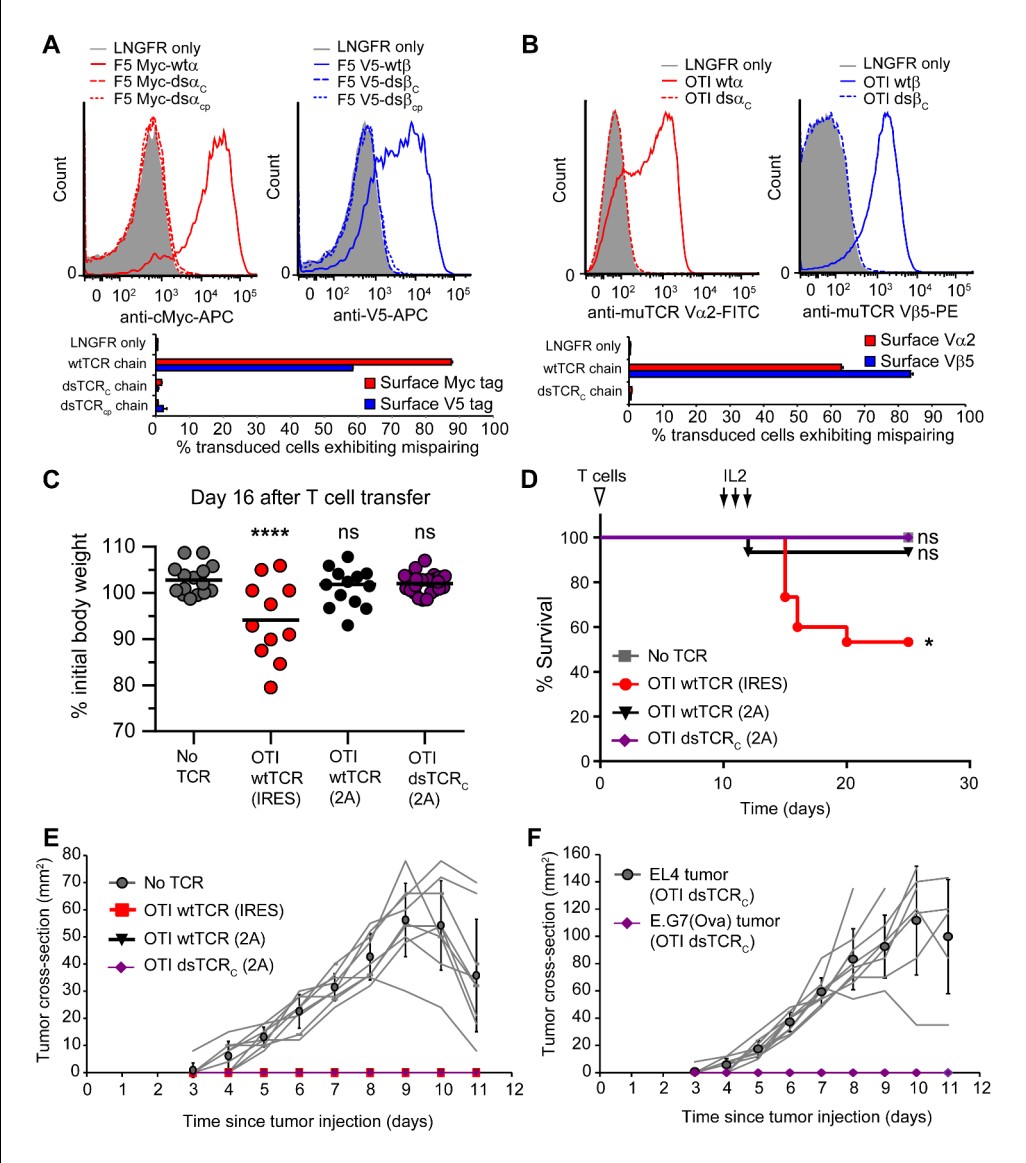

**Figure 4.** Domain-swapped TCRs do not mispair with endogenous TCRs and do not cause graft-vs-host disease. (A-B) Flow cytometric measurement of mispairing of (A) human F5 TCR chains in human peripheral blood T cells and (B) murine OTI TCR chains in murine splenic T cells. Cells were transduced with only TCRα (left panel) or only TCRβ (right panel) such that expression of the introduced chain on the cell surface required mispairing with endogenous TCR chains. Representative histograms are displayed above bar graphs quantifying means ± SD for 3 technical replicates. (C-D) Mouse model of TCR gene transfer-induced graft-vs-host disease (TI-GvHD). Mice were irradiated on day -1, administered TCR-transduced T cells on day 0, and injected twice daily with IL-2 from day 10–12. Weight was monitored daily and mice were euthanized at 85% initial body weight. Results are aggregated from 3 independent experiments. Respective group sizes, n = 9, 15, 15, and 25. *p<0.05, ****p<0.0001, ns, not significant. (C) Percent initial body weight of surviving mice at 16 days after T cell administration. (D) Kaplan-Meier survival curve. (E) Tumor size in mice administered TCR-transduced T cells and then injected with ovalbumin-expressing E.G7 thymoma tumor cells. Respective group sizes, n = 9, 6, 10, and 10. Mean ± SD for each group is overlaid on traces for individual animals. (F) Tumor size in mice (n = 10) administered OTI dsTCR-transduced T cells and then injected on opposing flanks with EL4 (control) or E.G7 (target) thymoma tumor cells.

The following figure supplements are available for figure 4:

**Figure supplement 1.** N-terminal epitope tags do not impair surface expression of TCR chains.

*Figure 4 continued on next page*

*Figure 4 continued*

**Figure supplement 2.** Wild-type TCR- and dsTCR-transduced T cells secrete a similar complement of cytokines upon stimulation.

**Figure supplement 3.** Domain-swapped TCR-transduced T cells are functional in vivo.

**Figure supplement 4.** Wild-type and domain-swapped TCR-transduced T cells similarly protect against tumor growth

The lack of TI-GvHD in mice receiving dsTCR-transduced T cells was not due to lack of function in vivo. Mouse T cells transduced with OTI wtTCR or dsTCR$_C$ produced a similar complement of cytokines upon stimulation with cognate peptide-MHC multimer (*Figure 4—figure supplement 2*). For at least 6 weeks following T cell injection, $V_\alpha2^+/V_\beta5^+$ T cells were present in the spleen and retained specificity for H-2K$^b$/Ova tetramer (*Figure 4—figure supplement 3A*). These splenic T cells exhibited antigen-specific IFN-γ release and modulated the surface expression of T cell activation markers upon stimulation with ovalbumin peptide ex vivo (*Figure 4—figure supplement 3B,C*). The ovalbumin-expressing murine thymoma line, E.G7, grew in all control mice injected with LNGFR-transduced T cells, but did not grow detectably in any mice injected with either wtTCR- or dsTCR-transduced T cells (*Figure 4E*). Tumor cell killing was antigen-specific, because EL4 – the ovalbumin-negative parental line from which E.G7 was derived – grew readily in mice injected with dsTCR-transduced T cells (*Figure 4F*). The relative efficacies of wtTCR- and dsTCR-transduced T cells were compared by titrating down the number of T cells injected prior to tumor cell injection. Both wtTCR- and dsTCR-transduced T cells slowed tumor growth when mice were injected with 1 T cell per 500 tumor cells, and prevented tumor growth entirely at higher tested ratios (*Figure 4—figure supplement 4*). Thus, dsTCRs mediate antigen-specific immunity but not deleterious autoimmunity in vivo.

## dsTCRs complement other strategies to improve efficacy and safety of TCR gene therapy

Swapping TCR domains prevents mispairing with endogenous TCR chains, but also reduces the amount of transduced TCR on the cell surface in the presence of competing endogenous TCR chains. Accordingly, employing dsTCRs in adjunct with approaches to reduce endogenous TCR expression may increase dsTCR surface expression while also improving the safety of these approaches in a complementary manner. We explored the compatibility of dsTCRs with two such approaches: knockdown of endogenous TCR in mature T cells, and hematopoietic stem cell TCR gene therapy.

A recent report demonstrated that RNAi-mediated silencing of endogenous TCR genes improves surface expression and reduces mispairing of a co-delivered codon-optimized TCR (*Bunse, 2014*). However, TI-GvHD was not eliminated entirely, indicating incomplete knockdown of endogenous TCR. We hypothesized that combining dsTCRs with endogenous TCR knockdown would improve cell surface expression of dsTCRs while also eliminating TI-GvHD entirely. To investigate this combination approach, we transduced mouse T cells with vectors that co-delivered 1) shRNA targeting endogenous TCR α and β genes and 2) codon-optimized wt- and dsTCR versions of the lymphocytic choriomeningitis virus (LCMV)-specific P14 murine TCR. As previously shown (*Bunse, 2014*), the β chain ($V_\beta8$) of P14 wtTCR was disproportionally detected on the cell surface relative to the α chain ($V_\alpha2$) - indicative of mispairing - even when translational stoichiometry was ensured through the use of a viral 2A sequence (*Figure 5A*). Co-delivery of shRNA resulted in knockdown of endogenous TCR surface expression to ~40% of normal levels (*Figure 5—figure supplement 1*), improving the balance of P14 wtTCR chains expressed on the surface and increasing tetramer staining (*Figure 5A, B*). Reflecting the lack of mispairing exhibited by dsTCRs, the domain-swapped α and β chains were detected in equal amounts on the cell surface with or without endogenous TCR knockdown (*Figure 5A*). However, knockdown greatly improved surface expression and tetramer binding of the P14 dsTCR by reducing competition for assembly with CD3 chains (*Figure 5B*).

Mice injected with P14 wtTCR-transduced T cells exhibited weight loss and developed TI-GvHD, which was prevented by co-delivery of shRNA (*Figure 5C–E*). By contrast, no TI-GvHD was observed

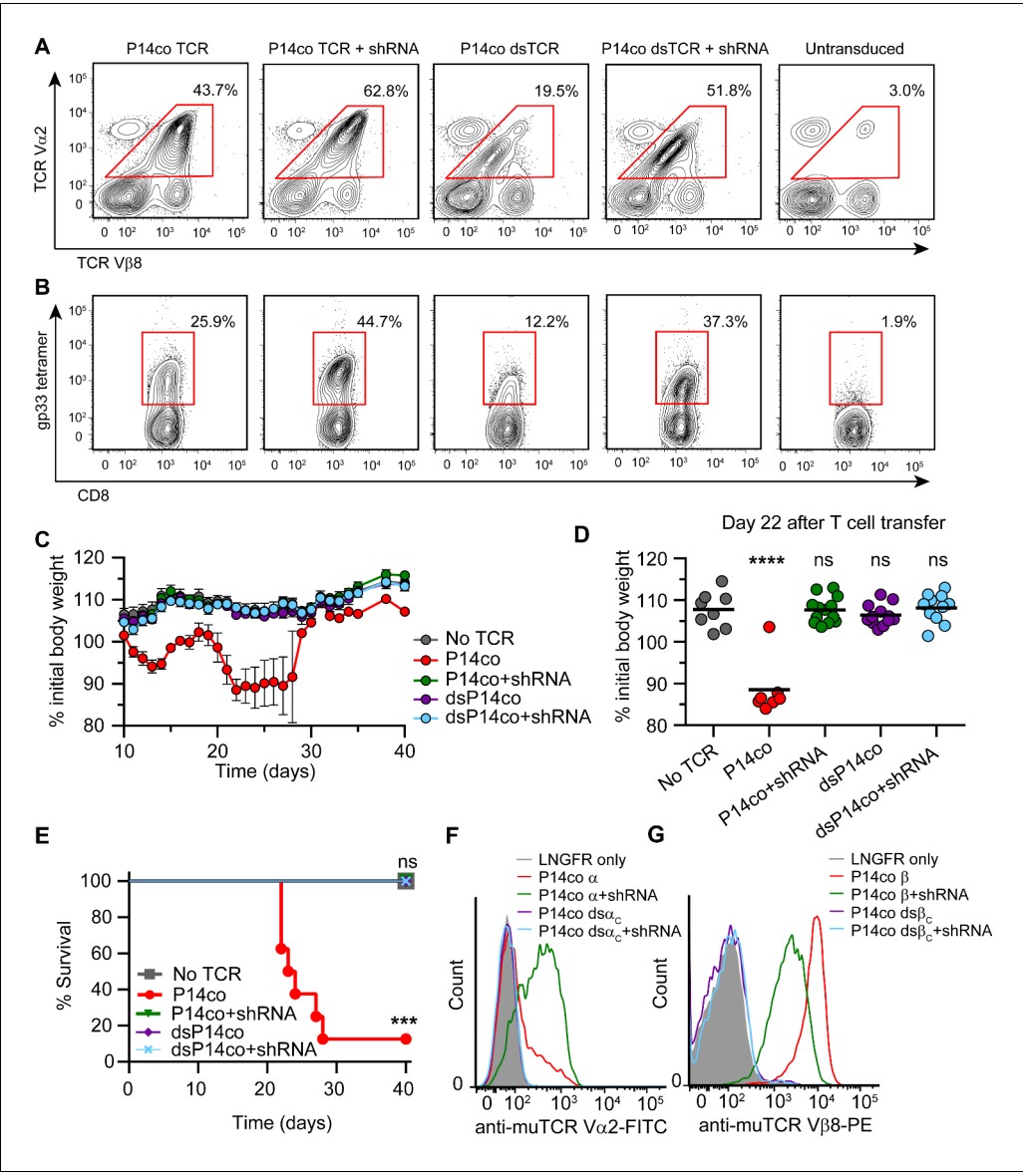

**Figure 5.** Complementarity of TCR domain-swapping and endogenous TCR knockdown. (**A-B**) Representative flow cytometry plots showing percent of CD8$^+$ T cells that are (**A**) V$_\alpha$2$^+$V$_\beta$8$^+$ and (**B**) gp33 tetramer$^+$ following transduction with a vector delivering codon-optimized gp33-specific P14 (V$_\alpha$2V$_\beta$8) TCR ± shRNA targeted to endogenous TCR C$_\alpha$/C$_\beta$. (**C-E**) Mouse model of TCR gene transfer-induced graft-vs-host disease (TI-GvHD), as in *Figure 4*, except mice were administered T cells transduced with codon-optimized P14 TCR ± shRNA targeted to endogenous TCR C$_\alpha$/C$_\beta$. Respective group sizes, n = 8, 8, 13, 12, and 11. ***p<0.001, ****p<0.0001 ns, not significant. (**C**) Mean percent initial body weight ± SD for each group over time. (**D**) Percent initial body weight of surviving mice at 22 days after T cell administration. (**E**) Kaplan-Meier survival curve. (**F-G**) Flow cytometric measurement of mispairing of (**F**) P14 TCRα and (**G**) P14 TCRβ chains in murine splenic T cells transduced with either chain ± shRNA targeted to endogenous TCR C$_\alpha$/C$_\beta$. Histograms are representative of triplicate results from 3 independent experiments.

The following figure supplement is available for figure 5:

**Figure supplement 1.** shRNA-mediated knockdown reduced endogenous TCR expression on surface of mouse T cells.

in mice receiving P14 dsTCR-transduced T cells, whether or not shRNAs were co-delivered. As with OTI dsTCR (*Figure 4B*), P14 dsTCR did not induce TI-GvHD because the P14 domain-swapped α and β chains did not mispair with endogenous chains (*Figure 5F,G*). By contrast, individually delivered P14 wild-type α and β chains mispaired with endogenous chains even when both endogenous chains were knocked down. In fact, knockdown of both endogenous chains increased mispairing of P14 wtα, suggesting the reduction of competing endogenous α enabled more mispairing of the weak P14 wtα with the remaining endogenous β chain. Prevention of TCR mispairing via endogenous TCR knockdown therefore relies on both the reduction of endogenous TCR and the equal (over)expression of the transduced TCR chains. Taken together, these results suggest that a combination approach of endogenous TCR knockdown and dsTCR transduction may optimize safety and efficacy over either approach alone.

Hematopoietic stem cells (HSCs) transduced with pre-rearranged TCR genes differentiate into antigen-specific T cells in the periphery (*Vatakis et al., 2011*; *Yang and Baltimore, 2005*). This is an attractive therapeutic approach as it provides a continuous supply of antigen-specific naïve T cells with optimal replicative potential (*Hinrichs et al., 2011*). Additionally, expression of a pre-rearranged TCR in HSCs suppresses rearrangement of the endogenous TCR in these cells via allelic exclusion (*Hinrichs et al., 2013*; *Vatakis, 2013*). As allelic exclusion in TCR-transduced HSCs is incomplete, however, there remains a risk that TCR mispairing can precipitate autoimmunity. HSCs transduced with dsTCRs would retain the advantages of wtTCR-engineered HSCs while eliminating the risk of autoimmunity. We injected HLA-A*0201$^+$ immunodeficient (NSG) mice with F5 wtTCR- or dsTCR-transduced human CD34$^+$ HSCs. Three months after injection, A2/MART1-specific CD3ε$^+$ T cells constituted up to 87.7% of human CD45$^+$ thymocytes and up to 66.7% of human CD45$^+$ CD19$^-$ CD3ε$^+$ splenocytes in mice engrafted with dsTCR-transduced stem cells (*Figure 6*). Therefore, dsTCRs are compatible with TCR-engineered HSC adoptive transfer.

## Substitution of αβ TCR constant domains with corresponding domains from γδ TCRs is a related strategy to prevent TCR mispairing

A subset of T cells encode γδ TCRs, which share a broadly similar architecture to αβ TCRs, are somatically rearranged, associate with CD3 chains, and mediate antigen-specific cellular immunity, but which are functionally distinct from αβ TCRs (*Attaf et al., 2015*). Interestingly, αβ TCR chains transduced into γδ T cells do not mispair with endogenous γδ TCR chains (*Saito, 1988*; *van der Veken et al., 2006*). These αβ TCR-transduced γδ T cells mediate antigen-specific immunity in vitro and in mouse models (*van der Veken et al., 2006*, *2009*). However, the relative paucity of γδ T cells in the blood, the requirement of co-delivery of CD4 and CD8 co-receptors, and the lower proliferative capacity and in vivo effector function of these cells relative to αβ T cells are challenges of using αβ TCR-transduced γδ T cells for adoptive cell therapies (*van der Veken et al., 2006*, *2009*).

Ideally, the lack of mispairing between αβ and γδ TCRs could be exploited in the context of the more abundant and co-receptor-expressing αβ T cells. We explored swapping corresponding constant domains of αβ and γδ TCRs as an approach to prevent mispairing that is conceptually related to but distinct from our αβ interchain domain-swapping approach. We designed three chimeric constructs in which the F5 TCR domains V$_α$ and V$_β$ were fused to C$_δ$ and C$_γ$ (dsTCR$_{Cδγ}$); V$_α$C$_α$ and V$_β$C$_β$ were fused to cp$_δ$ and cp$_γ$ (dsTCR$_{δγ}$); or V$_α$C$_α$ and V$_β$C$_β$ were fused to cp$_γ$ and cp$_δ$ (dsTCR$_{γδ}$), (*Figure 7A*). Mispairing between these constructs and wild-type endogenous αβ TCR chains is expected to be disfavored due to juxtaposition of αβ and γδ constant domains (*Figure 7B*).

Constructs were initially tested for surface expression and dextramer binding in transduced Jurkat T cells. As with the interchain domain-swapped constructs (*Figure 2B*), chimeric domain-swapping at the V-C junction resulted in loss of TCR surface expression (*Figure 7C*). By contrast, both dsTCR$_{δγ}$ and dsTCR$_{γδ}$ expressed on the surface and bound cognate dextramer to a similar extent as wtTCR. As expected, 3 of the 4 simulated mispaired constructs comprising one chimeric chain and one wtTCR chain did not bind dextramer (*Figure 7D,E*). Only the wtα/dsβ$_γ$ mispaired construct was detected at a low level on the cell surface, binding dextramer at 20% the level of wtTCR (*Figure 7D*). Dextramer binding results were similar in primary T cells, indicating that chimeric dsTCR$_{δγ}$ and dsTCR$_{γδ}$ assemble with CD3 chains with similar efficiency as wtTCR in the presence of competing endogenous TCR (*Figure 7F*; MFI for dsTCR$_{δγ}$ and dsTCR$_{γδ}$ = 157% and 174% that of wtTCR, respectively). T cells expressing chimeric dsTCRs were activated and secreted IFN-γ in response to coincubation with cognate antigen-expressing target cells (*Figure 7G,H*). In contrast to

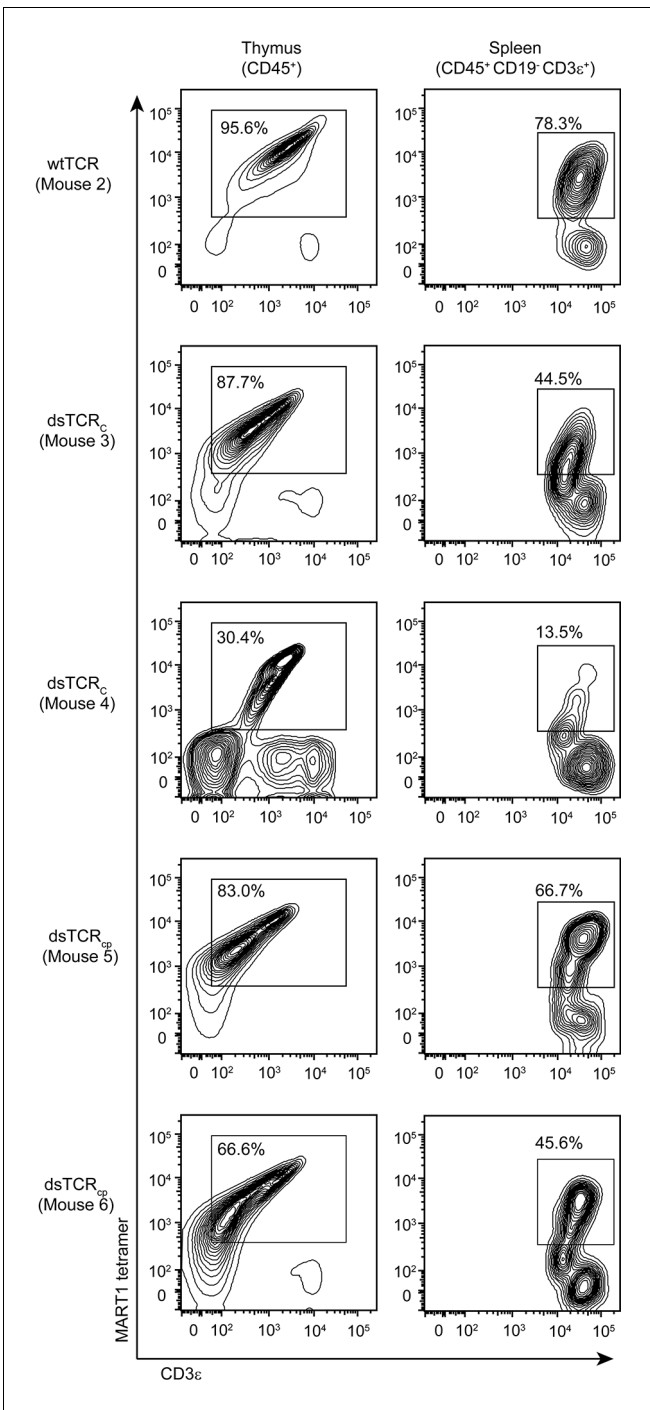

**Figure 6.** dsTCR-transduced human HSCs develop into antigen-specific T cells in humanized mice. Flow cytometry plots showing dissociated thymocytes or splenocytes from humanized mice stained with A2/MART1 tetramer and anti-CD3ε. Mice were injected with TCR-transduced HSCs three months prior to analysis. Mouse 3 and 4 are replicates, mouse 5 and 6 are replicates. Mouse 2 has no replicate because wtTCR-transduced HSCs did not engraft in Mouse 1 (not shown). Results are from a single experiment.

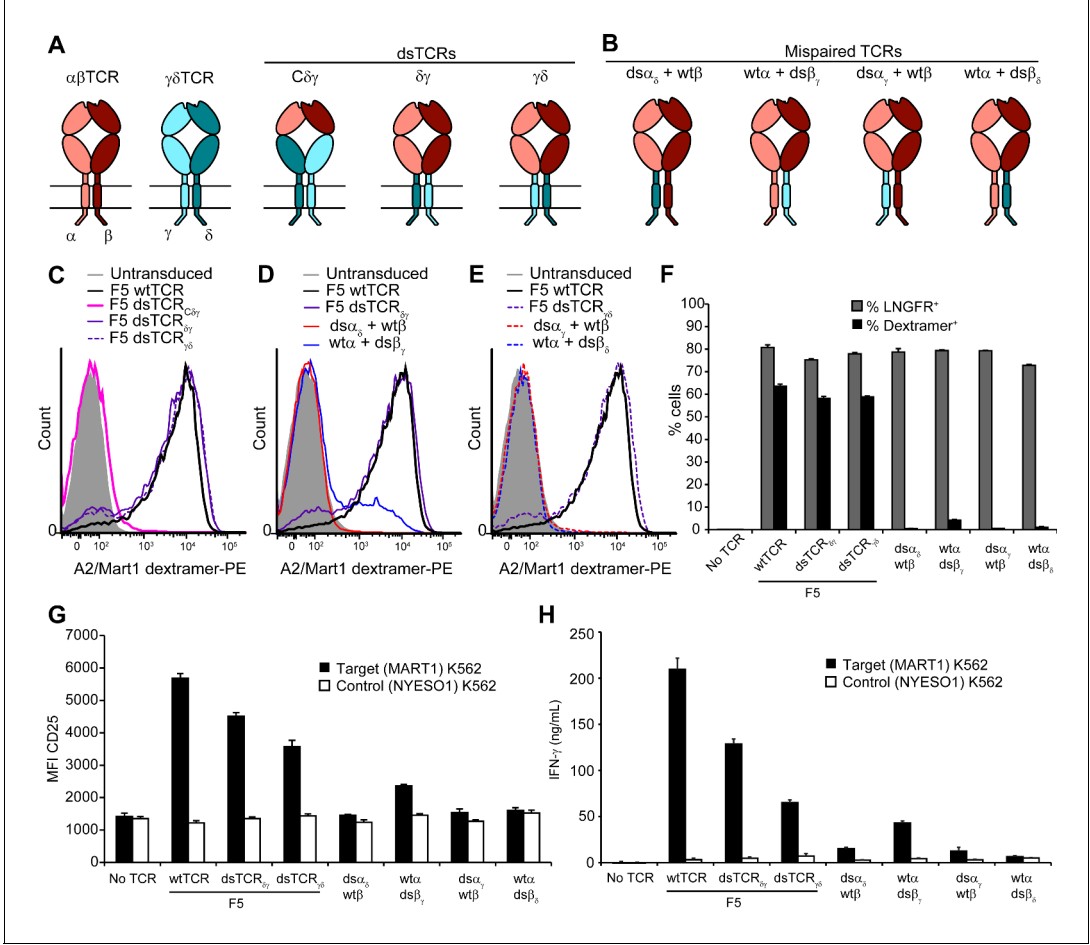

**Figure 7.** Substitution of αβ TCR constant domains with corresponding domains from γδ TCRs prevents TCR mispairing. (**A**) Schematic of αβ/γδ chimeric TCR architectures. Domains are swapped following the V domains (dsTCR$_{Cδγ}$) or C domains (dsTCR$_{δγ}$ and dsTCR$_{γδ}$). (**B**) Schematic of simulated mispaired constructs, in which αβ and γδ constant domains are juxtaposed. Such mispairing is expected to be unproductive because αβ and γδ constant domains do not interact. (**C-E**) Flow cytometry histograms comparing peptide-MHC multimer binding by Jurkat T cells transduced with (**C**) F5 wtTCR or αβ/γδ chimeric dsTCR constructs, (**D**) F5 wtTCR, F5 dsTCR$_{δγ}$ and constructs simulating mispairing between these, or (**E**) F5 wtTCR, F5 dsTCR$_{γδ}$ and constructs simulating mispairing between these. Histograms are representative of triplicate results from 2 independent experiments. (**F**) Flow cytometric measurement of percent of dextramer-binding TCR and LNGFR expressed on the surface of transduced primary human T cells. LNGFR is an independent transduction marker expressed from the same vector as TCR. (**G**) Flow cytometric measurement of CD25 expressed on the T cell surface and (**H**) ELISA measurement of secreted IFN-γ from TCR-transduced primary human T cells following 48 hr coincubation with cognate or control antigen-expressing K562 target cells. For panels (**F-H**), means ± SD for 3 technical replicates are shown.

The following figure supplement is available for figure 7:

**Figure supplement 1.** Chimeric αβ/γδ dsTCRs function similarly with cytoplasmic domains derived from αβ or γδ TCRs.

dextramer binding, these functional outputs were reduced for chimeric dsTCRs relative to wtTCR, indicating chimeric domain-swapping partially impaired signaling capacity. Replacing the chimeric TCR cytoplasmic signaling domains with those from αβ TCR did not substantively alter the function of chimeric dsTCRs (*Figure 7—figure supplement 1*). As observed in Jurkats, chimeric dsα$_δ$, dsα$_γ$, and dsβ$_δ$ chains did not mispair productively with wild-type αβ TCR chains (*Figure 7F–H*). Only the wtα/dsβ$_γ$ mispaired construct exhibited low levels of dextramer binding (7% of wt) and secreted IFN-γ (20% of wt), indicating some allowance for pairing between constant TCRα and TCRγ domains. Thus, substitution of constant domains of tumor-specific αβ TCRs with corresponding γδ TCR domains produces a chimeric receptor that mediates functional, antigen-specific T cell immunity with minimized mispairing risk.

## Discussion

The promise of TCR gene transfer – to enable us to initiate or amplify beneficial immune responses – is already coming to fruition in the field of cancer immunotherapy (*Johnson et al., 2009*; *Morgan, 2006*; *Davis, 2010*; *Parkhurst, 2011*; *Robbins, 2015, 2011*). However, mispairing between introduced and endogenous TCR chains produces autoreactive human T cells *in vitro* (*van Loenen et al., 2010*) and causes graft-vs-host disease in mice (*Bendle et al., 2010*; *Bunse, 2014*), indicating that TCR gene transfer poses a non-theoretical risk of autoimmunity. As the potency of TCR gene therapy is augmented through combination with other immunotherapies (e.g. checkpoint blockade), there may be a concomitant rise in mispairing-related serious adverse events. This work describes a domain-swapping strategy to prevent TCR mispairing, thereby improving the safety of this promising approach.

By swapping constant domains between α and β chains of a clinically-relevant αβ TCR, we generated a receptor that retains antigenic specificity and function but does not mispair with endogenous TCRs. Interchain domain-swapped TCRs prevent mispairing-related autoreactivity through three potential mechanisms: (1) dsTCR chains mispair with wtTCR chains inefficiently due to juxtaposition of non-interacting domains (i.e. neither α nor β self-associate); (2) mispairing between wt and ds chains results in a receptor that incompletely assembles with CD3 chains and is degraded; (3) mispaired wt/dsTCRs that escape degradation and express on the surface with an incomplete complement of CD3 chains will have impaired signaling capacity. Thus, proper dsTCR/CD3 complex assembly is central to the differential function of correctly paired versus mispaired dsTCRs. The detailed mechanism by which dsTCRs assemble with CD3 chains remains open. The transmembrane domains and connecting peptides of wtTCRs participate in complex assembly while the ectodomains contribute to complex stability and signaling (*Call et al., 2002*; *Kuhns and Davis, 2007*; *Xu and Call, 2006*). The dsTCRs reorient these domains with respect to one another. Despite this, dsTCRs assemble with CD3 chains with proper juxtaposition of CD3ε chains, express on the cell surface, and mediate functional, antigen-specific responses, including tumor cell killing. This indicates that dsTCR/CD3 complexes preserve the transmembrane contacts between the TCR and CD3 chains that are essential for complex assembly (*Call et al., 2002*). If transmembrane orientation is preserved, it is unlikely that the short, rigid CD3 connecting peptides can contact the reoriented TCR ectodomains, resulting in presumed (though undemonstrated) disruption of these stabilizing contacts. This could account for the generally lower surface expression observed for dsTCRs relative to the wtTCR. The dsTCR$_{cp}$ construct additionally reorients the connecting peptides with respect to the transmembrane domains, consistent with its generally lower surface expression relative to the dsTCR$_C$ construct. That both of these constructs transduce extracellular ligand binding into intracellular signaling suggests that the function of the TCR/CD3 complex is remarkably robust toward structural perturbation.

The functional avidity of dsTCR-transduced T cells can be optimized for clinical application in several ways. First, wtTCR- and dsTCR-transduced Jurkat T cells performed equivalently in functional assays, indicating the importance of high transduction efficiency. Second, assembly of dsTCRs with CD3 may be improved by reducing the number of endogenous TCRs competing for these chains. In this work, we showed that dsTCRs are compatible with TCR-transduced HSC transfer, as well as with shRNA-mediated knockdown of endogenous TCR genes. Combining dsTCRs with these adjunct approaches improved cell surface expression of dsTCRs and safeguarded against mispairing with residual endogenous TCR. Third, swapping domains closer to the V-C junction would avoid reorienting constant domains with respect to each other, thereby preserving all contacts with CD3. We attempted to optimize such a construct in this work (dsTCR$_V$ and dsTCR$_{Cδγ}$), but found that all constructs that swapped domains at the V-C junction abrogated TCR surface expression completely. Interestingly, a conceptually related approach has been used to ensure correct pairing of heavy and light chains in bispecific antibodies (*Schaefer et al., 2011*), and the lead architecture (CrossMab$^{CH1-CL}$) from that application swapped domains at the V-C junction (i.e. the elbow). It is unclear why the dsTCR$_V$ variant was entirely nonfunctional whereas analogously manipulated CrossMab antibodies bind antigen with affinity comparable to wild-type Fab. This may be attributable to a more flexible antibody hinge or to the specifics of how the domain-swap was achieved. Notably, the CrossMab approach differed from ours in that domains were swapped asymmetrically between chains (i.e. heavy chain residues were duplicated in the hinge/elbow region of both the heavy and light chains),

so a similar modification to our approach may identify a functional dsTCR design that is swapped near the V-C junction.

Finally, although we focus on TCRs relevant to cancer immunotherapy, domain-swapping was applied successfully to all TCRs attempted, including four human TCRs and two mouse TCRs. Thus, the general dsTCR approach can be applied broadly to TCRs of therapeutic interest, enabling development of safer immunotherapies for a variety of cancers, infections, and autoimmune diseases.

## Materials and methods

### Materials

Primers were purchased from Integrated DNA Technologies (Coralville, IA). KOD polymerase master mix and polybrene were purchased from EMD Millipore (Darmstadt, Germany). Sequencing was performed by Retrogen Inc (San Diego, CA). Fluorescently labeled antibodies and 7-AAD for flow cytometry were purchased from Biolegend (San Diego, CA), eBioscience (San Diego, CA), or Beckman Coulter (Brea, CA). Peptides were purchased from Anaspec Inc (Fremont, CA) or Thermo Fisher Scientific (Waltham, MA). Anti-CD3 (OKT3) and anti-CD28 (CD28.2) activating antibodies were purchased from eBioscience (San Diego, CA). Short-dated Proleukin was provided by Prometheus Laboratories Inc (San Diego, CA) through an investigator-initiated trial program. All other cytokines were purchased from Peprotech, Inc. (Rocky Hill, NJ). Concanavalin A was purchased from Sigma-Aldrich (St. Louis, MO). RetroNectin was purchased from Clontech Laboratories (Mountain View, CA). BioT transfection reagent was purchased from Bioland Scientific (Paramount, CA). Cell culture media, antibiotics, and fetal bovine serum were purchased from Corning (Corning, NY). Human AB serum was purchased from Omega Scientific (Tarzana, CA). Peptides were purchased from Anaspec (San Jose, CA).

### Cell lines

293T/17, Jurkat E6-1, and K562 cells were purchased from the American Type Culture Collection (Manassas, VA). Their identities were authenticated by STR analysis and they were confirmed to be mycoplasma-free (DDC Medical, Fairfield, OH). 293T cells were grown in D10 media (Dulbecco's Modified Eagle Medium (DMEM) supplemented with antibiotics (penicillin/streptomycin) and 10% (v/v) fetal bovine serum (FBS)). Jurkat and K562 cells were grown in C10 media ((RPMI 1640) medium supplemented with antibiotics, 10% (v/v) FBS, 10 mM HEPES, 50 µM β-mercaptoethanol, 1x MEM NEAA, and 1 mM sodium pyruvate). EL-4 and E.G7 mouse thymoma cell lines were provided by Lili Yang (UCLA) and grown in C10 media. The cells were split every 2–3 days. All cells were grown and assayed at 37°C with 5% atmospheric $CO_2$.

### Animals

Female 6 wk old C57BL/6 J (B6) mice and B6.SJL-$Ptprc^aPepc^b$/BoyJ (CD45.1 B6) mice were purchased from Jackson Laboratory (Bar Harbor, ME) and housed in the California Institute of Technology animal facility. Animals were granted access to food (PicoLab Rodent Diet 5053, PMI Nutrition International, St. Louis, MO) and water ad libitum throughout the study. The room was maintained on a 13:11 hr light:dark cycle. The ambient temperature remained between 71–75 °F with a relative humidity of 30-70%. All injections (retroorbital, intraperitoneal, and subcutaneous) were performed under anesthesia (isoflurane). Local anesthetic (0.5% proparacaine hydrochloride) was administered following retro-orbital injections. Animals were sacrificed by controlled $CO_2$ inhalation. Experiments conducted at Caltech were approved by the Institutional Animal Care and Use Committee of the California Institute of technology (IACUC protocol 1611).

Animals for TCR/shRNA combination experiments at MDC Berlin were purchased as 6 wk old female C57BL/6 (Charles River (Sulzfeld, Germany) and housed at the MDC SPF facility. These experiments were conducted in accordance with institutional and national guidelines, after approval by a state animal ethics committee (Landesamt für Gesundheit und Soziales protocol 0233/11). Animals for HSC transduction experiment were bred, housed, and monitored according to UCLA Department of Laboratory Animal Medicine standards (UCLA Animal Research Committee protocol 2008–167).

## Peptide-MHC multimers

HLA-A2 and β2-microglobulin were expressed in *E. coli*, refolded in the presence of MART1$_{26-35}$ $_{(A27L)}$ (ELAGIGILTV) or NY-ESO-1$_{157-165(C165V)}$ (SLLMWITQV) heteroclitic peptides, purified, and biotinylated as previously described (*Toebes et al., 2006*). Biotinylated H-2K$^b$/Ova$_{257-264}$ monomers were provided by the NIH Tetramer Core Facility (Atlanta, GA). Peptide-MHC monomers were stored at −20°C or −80°C until use. Peptide-MHC tetramers were prepared from monomers using fluorescently-labeled streptavidin from Molecular Probes (Eugene, OR) according to the NIH Tetramer Core Facility protocol. Peptide-MHC dextramers were prepared by doping the tetramer assembly reaction with biotinylated dextran (manuscript in preparation). Briefly, peptide-MHC and fluorescently-labeled streptavidin were added at a molar ratio of 3:1 (1.5 µM:0.5 µM) and incubated 10 min. Biotinlyated dextran, MW 500 kDa (Molecular Probes) was then added to the incubation at 25 nM (1:20 with respect to streptavidin) and incubated an additional 10 min. Peptide-MHC multimers were prepared fresh or stored briefly at 4°C.

## TCR, CD3, and pMHC single-chain trimer constructs

OTI TCR genes were obtained from Gavin Bendle (Univ. of Birmingham, UK). F5 TCR genes were obtained from Steve Rosenberg (NCI, Bethesda, MD). 1 G4 TCR genes were obtained from Richard Koya (RPCI, Buffalo, NY). M1 TCR genes and mouse and human CD3 genes were obtained from Lili Yang (UCLA, Los Angeles, CA). Novel TCR architectures were prepared by assembly PCR. Amino acid sequences of hybrid joints are provided within relevant figures. For epitope-tagged TCR constructs, Myc or V5 epitope tags were inserted between the leader sequence and the N-terminus of the TCR chain. Peptide-MHC single-chain trimers of A2/NY-ESO-1(SLLMWITQV) and A2/MART1 (ELAGIGILTV) were prepared with a disulfide trap modification as described (*Hansen et al., 2009*).

## Evaluation of TCR surface expression in transfected 293T

293T were plated (25 K/well on 96-well plate) a day prior to transfection. Cells in each well were transfected with 0.2 µg each of plasmids encoding TCR, CD3δ, CD3ε, CD3γ, and CD3ζ, using BioT transfection reagent according to manufacturer's instructions. Media was refreshed after 24 hr. Cells were de-adhered from the plate using 2 mM EDTA in PBS, stained with fluorescent antibodies and pMHC multimers in FACS buffer (2% FBS in PBS), and analyzed by flow cytometry 48 hr after transfection.

## Functional assays in jurkat T cells

TCR constructs were subcloned into a pCCLc-MND-based lentiviral vector in the format TCRα-F2A-TCRβ-P2A-Myc271. Myc271 is a novel chimeric transduction marker comprising the transmembrane and truncated extracellular domains of CD271(LNGFR) fused to an extracellular cMyc epitope tag. Lentiviral vectors were prepared from 293T producer cells. One day prior to transfection, 1 x 10$^6$ 293T cells/well were plated on 6-well plates. In 100 µL PBS, added 1 µg TCR lentivector, 1 µg pCMV-R8.2 (gag-pol helper plasmid), 0.2 µg pCAGGS-VSVG (pseudotyping helper plasmid), and 6.6 µL polyethyleneimine transfection reagent (Sigma) and incubated for 10 min. Fresh 2 mL media was replaced on plated 293T, and then transfection mix was added to cells. 48 hr following transfection, Jurkat T cells (10 (*Johnson et al., 2009*) cells/well in 250 µLC10) were added on a 24-well plate and 250 µL filtered (0.45 µm) viral supernatant was added to cells to initiate transduction. After 60–72 hr of transduction, Jurkats were assayed for TCR surface expression and function. To test for surface expression, cells were rinsed and stained with fluorescent antibodies and pMHC multimers in FACS buffer, and analyzed by flow cytometry. Dead cells were excluded from analysis using 7-AAD. To test for function, unselected Jurkat T cells were co-incubated in a 96-well plate at a 1:1 ratio with K562 cells stably expressing control or cognate peptide-MHC single chain trimers. After 48 hr of coincubation, the supernatant was tested for secreted interleukin-2 with a commercial ELISA kit according to the manufacturer's protocol (BD, Franklin Lakes, NJ).

## Functional assays in primary human T cells

TCR constructs were subcloned into an MSCV-based retroviral vector in the format LNGFRΔ-P2A-TCRα-F2A-TCRβ. LNGFRΔ is a transduction marker comprising the low-affinity nerve growth factor receptor with intracellular domain truncated. Retroviral vectors were prepared from 293T producer

cells. One day prior to transfection, 6-well plates were coated with poly-L-lysine and 1 x $10^6$ 293T cells/well were plated. On day of transfection, 2 μg TCR lentivector, 2 μg pHIT60 (gag-pol helper plasmid), and 1.3 μg pRD114 (pseudotyping helper plasmid) were mixed in 100 μL 250 mM $CaCl_2$. DNA/$CaCl_2$ was added while vortexing to 100 μL 2x HEPES-buffered saline (HBS), incubated 5 min, and then added to cells in fresh D10 media. The day after transfection, media was replaced with D10 containing 20 mM HEPES and 10 mM sodium butyrate, incubated 8 hr, and then media was replaced with D10 containing 20 mM HEPES. Viral supernatant was harvested 40–48 hr after transfection, filtered (0.45 μm), and used directly to transduce activated T cells.

Primary human peripheral blood mononuclear cells (PBMCs) were purchased from the CFAR Virology Core Lab at the UCLA AIDS Institute. Two days prior to transduction, a 24-well plate was coated with 1 μg/mL anti-CD3 (OKT3) and peripheral blood mononuclear cells were activated at 2 x $10^6$ cells/mL on the anti-CD3 coated plate in the presence of 1 μg/mL anti-CD28 and 300 U/mL IL-2 in T cell media (AIM-V media supplemented with 5% (v/v) human AB serum and antibiotics (pen/strep)). On day of transduction, activated T cells were centrifuged with virus and 10 μg/mL polybrene at 1350xg for 90 min at 30°C. Following spinfection, virus was replaced with fresh T cell media supplemented with 1 μg/mL anti-CD28 and 300 U/mL IL-2. 48 hr following transduction, transduced primary T cells were assayed for TCR surface expression. Functional assays were performed with unselected transduced T cells as described for Jurkat T cells, except that after coincubation with antigen-expressing K562 derivatives, surface expression of CD25 was measured for primary T cells and the supernatant was tested with a commercial ELISA kit for secreted interferon-γ (BD, Franklin Lakes, NJ).

## BaF3 proliferation assays

BaF3 cell lines were generated and analyzed by flow cytometry as previously described (*Kuhns et al., 2010*; *Lee et al., 2015*). Briefly, Phoenix E packaging cells were used to generate retrovirus for each cell line. In 6 cm plates, 3 μg of the desired retroviral construct along with 0.75 μg gag polymerase and 0.75 μg eco envelope were mixed in DMEM with 10% FCS and then transfected into 1 x $10^6$ Phoenix E packing cells using Turbofect (Fermentas) according to the manufacturer's instructions. The next day, transfected Phoenix E cells were shifted to 32°C after media change. Viral supernatants were harvested at 48 and 72 hr post-transfection. The supernatants for all constructs used to generate a cell line were pooled and concentrated to 250 μl using an Amicon Ultra 15 100 kDa (Millipore). BaF3 cells were co-transduced simultaneously with a polycistronic vector encoding all CD3 chains (cytoplasmic domains truncated and CD3ε fused to EpoR cytoplasmic domain) and a vector encoding a TCR construct. Parental BaF3 cells (1 x $10^6$) were plated in 2 ml of complete RPMI in one well of a 12 well plate in 8 μg/ml polybrene, 1 ng/μL IL-3 (R and D Systems), and the viral supernatant. Cells were then spun for 2 hr at 32°C at 2700 rpm in a Legend XTR centrifuge (Thermo-Fisher), after which the media was exchanged with fresh complete RPMI containing 1 ng/μL IL-3. Transduced cells were cultured overnight at 37°C prior to selection for εE expression with 10 μg/mL puromycin (InvivoGen). Cells transduced with the 2B4 control TCR were selected with 10 μg/mL puromycin and 100 μg/mL zeocin (LifeTech). BaF3 cells were split as necessary to keep them sparse and under heavy drug selection. All cell lines were maintained at a density below 1 x $10^6$/mL and split the night before experiments to be in mid growth phase the day of the experiment.

Surface expression was assessed by flow cytometry with the following antibodies: anti-CD3ε-PE-Cy7 (clone 2 C11), anti-Vα2-PE (clone B20.1), anti-Vβ5-FITC (clone MR9-4), and HLA-A2/MART1(ELA-GIGILTV) and H-2K$^b$/ovalbumin(SIINFEKL) dextramers. BaF3 cells (2.5 X $10^4$/well) were setup in 96 well plates in triplicate under drug selection for a 3-day proliferation assay in the absence of IL-3. Cells were enumerated by flow cytometry with count beads, which allowed the total cell number per well to be extrapolated.

## Adoptive transfer of transduced hematopoietic stem cells

Humanized mice were generated as previously described (*Gschweng et al., 2014*). Briefly, HLA-A*0201$^+$, CD34-enriched human peripheral blood stem cells were stimulated for 24 hr before transduction with spin concentrated virus encoding the indicated constructs at a vector concentration of 1 x $10^8$ transduction units/mL. Twenty four hours later, cells were collected, washed, and resuspended at a concentration of 2 x $10^7$ cells/mL in X-VIVO-15 medium (Lonza). Neonatal (d3-5) NSG-

A0201 mice (Jax 014570) received 1 Gy from a [137]Cs source immediately before injection. A dose of 1 x 10[6] cells (50 µl) was delivered to each mouse via intrahepatic injection. Three months post-transplant, mice were harvested for analysis. Thymus and spleen from each mouse was surgically excised and dissociated over 70 µm strainers in RPMI + 10% FBS. Flow cytometry analysis was performed on 1 x 10[6] cells per sample, and stained with anti-mouse CD45, HLA-A2/MART1(ELAGIGILTV) multimer, and the following anti-human antibodies: CD45, CD19, CD3, CD4, and CD8. Data were acquired on an LSRFortessa (BD).

## Mouse T cell transduction with OTI TCR constructs

Murine OTI TCR constructs were subcloned into the MSCV-based retroviral vector pMX in the format TCRβ-(F2A or IRES)-TCRα. Retroviral vectors were prepared from 293T producer cells. Two days prior to transfection, 5 x 10[6] 293T cells were seeded per 15 cm plate. On day of transfection, 11.25 µg retroviral vector (encoding OTI wtTCR, OTI dsTCR, or LNGFRΔ control) and 11.25 µg pCL-Eco ecotropic packaging vector were mixed in 1000 µL serum-free DMEM. Transfection reagent (33.75 µL BioT) was added and transfection mix was immediately vortexed, incubated for 5 min, and then added to cells in fresh D10 media. The following day, media was replaced on 293T and a Retronectin-coated plate was prepared (24-well plate coated overnight with 30 µg/mL Retronectin overnight at 4°C and then blocked with 2% (w/v) BSA for 30 min at room temperature). Viral supernatant was harvested at 48 and 72 hr after transfection, filtered (0.45 µm), and added directly to blocked Retronectin-coated plates. Virus was bound to plate by spinning at 3000 xg for 120 min at 4°C. Supernatant was removed immediately prior to adding activated T cells to virus-coated plate.

Primary mouse T cells were harvested from mouse spleen. Mouse spleens were dissociated and treated with RBC lysis buffer (Biolegend) to remove erythrocytes. Mouse splenocytes (2 x 10[6] cells/well on a 24-well plate) were activated in C10 containing 2 µg/mL concanavalin A and 1 ng/mL IL-7 for 24 hr. Activated T cells were pooled, supplemented with 4 µg/mL protamine sulfate, spun on the virus-coated plate at 800 xg for 30 min at 32°C, and then incubated in the presence of virus overnight at 37°C. The next day, cells were washed with PBS, and then transduced a second time on fresh, virus-coated plates. The day after the second transduction, cells were counted and analyzed for transgene expression by flow cytometry. TCR-transduced cells were evaluated directly for activity or transferred to recipient mice for in vivo assays as described below.

## Ensemble cytokine secretion detection

Mouse splenocytes (2 x 10[6] cells/well) were activated in 1 mL C10 containing 2 µg/mL concanavalin A and 1 ng/mL IL-7 for 48 hr, then transduced with retroviral supernatant encoding OTI wtTCR, OTI dsTCR, or no TCR as described. After overnight transduction, T cells were washed and incubated for 16 hr in either C10 media alone or media containing 1:100 H-2K[b]/Ova tetramer (final 39 nM) and 1 µg/mL anti-CD28 (clone 37.51). Supernatants were collected after the 48 hr ConA/IL-7 activation and after the 16 hr tetramer stimulation for ensemble proteomic analysis based on a DNA-encoded Antibody Library (DEAL) assay (*Bailey et al., 2007*). Briefly, a DNA barcoded slide was sectioned into wells using molded elastomers. The surface of the slide was blocked for 1 hr with 3% bovine serum albumin (BSA, Sigma) in PBS buffer (Irvine Scientific). A cocktail of antibody-DNA conjugates that capture the measured proteins was added and incubated for an hour in order to hybridize them to the surface (*Bailey et al., 2007*; *Ma et al., 2011*). After washing the wells 3 times with 3% BSA, the supernatant containing the secreted proteins was added to individual wells and incubated for an hour. Afterwards, wells were washed again 3 times with 3% BSA. The assay was completed by applying biotinylated antibodies and streptavidin-Cy5 and a final wash with 3% BSA to remove excess dye. Finally, the slide was washed with PBS before spin drying and scanning on a GenePix 4400A fluorescent scanner (Molecular Devices).

## Evaluation of TCR gene transfer-induced graft-vs-host disease (TI-GvHD) in mice

The capacity of OTI wtTCR- and dsTCR-transduced T cells to induce graft-vs-host disease was evaluated by an assay simulating TCR gene therapy in mice (*Bendle et al., 2010*). Donor mouse (CD45.1 B6) splenocytes were activated and transduced with retroviral supernatant encoding OTI wtTCR, OTI dsTCR, or no TCR as described above. Recipient mice received 5 Gy from a [137]Cs source to render

them lymphopenic one day prior to T cell injection (d-1). The next day (d0), a dose of 1 x $10^6$ transduced cells (100 µl) was delivered to each mouse via retro-orbital injection. Initial body weight was recorded on d10, after which mice received twice daily intraperitoneal injections of 7.2 x $10^5$ U IL-2 (Prometheus Labs) for three days (d10-d12). Animals were monitored thereafter for signs of TI-GvHD (i.e. cachexia), and mice were euthanized if body weight fell to 85% initial weight.

## Ex vivo assay for splenic T cell activity

Donor mouse splenocytes were activated with ConA/IL-7, transduced with OTI wtTCR, dsTCR, or control vector, and then injected retro-orbitally into lymphopenic recipient mice as described above. Forty-four days following T cell injection, recipient mice were sacrificed and splenocytes were dissociated, RBC-lysed, and seeded in a 48-well plate at $10^6$ cells/mL in C10 alone or C10 containing 1 µg/mL ovalbumin$_{257-264}$ (SIINFEKL) peptide. After 72 hr incubation, the supernatant was tested with a commercial ELISA kit for secreted mouse interferon-γ (BD) and the T cells were tested by flow cytometry for surface expression of $V_\alpha2$ TCRα, $V_\beta5$ TCRβ, H-2K$^b$/Ova$_{257-264}$ tetramer-specific TCR, and activation markers (anti-CD25 (clone PC61), anti-CD44 (clone IM7), and CD62L [clone MEL-14]).

## Tumor xenograft experiments

A subset of mice evaluated for TCR gene transfer-induced GvHD were further tested for capacity of transferred T cells to prevent tumor growth of tumor xenografts. Recipient mice were rendered lymphopenic by γ-irradiation, injected with TCR-transduced T cells retro-orbitally, and monitored for TI-GvHD as described above. Forty days following T cell injection, mice were injected on their right flank subcutaneously with 5 x $10^6$ ovalbumin-expressing E.G7 thymoma tumor cells. For tumor specificity experiment, mice were injected with OTI dsTCR-transduced T cells as above and 25 days thereafter injected subcutaneously with 5 x $10^6$ E.G7 (target) thymoma tumor cells on the right flank and 5 x $10^6$ EL4 (control) thymoma tumor cells on the left flank. For T cell titration experiment, mice were injected as above with unsorted splenocytes containing the specified number of transduced ($V_\alpha2^+V_\beta5^+$) CD8$^+$ T cells and 25 days thereafter injected with 5 x $10^6$ E.G7 thymoma tumor cells. The two longest perpendicular dimensions of each injected tumor were measured with calipers daily and used to calculate the tumor cross-sectional area. Mice were euthanized if the longest dimension measured 15 mm.

## Evaluation of safety of dsTCR/shRNA combination therapy in mice

Retroviral vectors encoding codon-optimized TCR and shRNA targeting endogenous TCR α and β genes were designed and produced as described (*Bunse, 2014*). Briefly, ecotropic virus particles were produced using Platinum-E packaging cells and γ-retroviral MP71 vector plasmids. Supernatants were harvested 48 hr after transfection, filtered (0.45 µm) and frozen (−80°C). Mouse splenocytes were activated using concanavalin A (2 µg/ml, Sigma-Aldrich) and IL-7 (1 ng/ml, Peprotech) for 24 hr and then the first round of transduction was performed. The cells were transferred into virus-coated plates, protamine sulfate (4 µg/mL, Sigma-Aldrich) was added and the plates were centrifuged (800 g, 32°C) for 30 min. The virus-coated plates were prepared by centrifugation of Retronectin-coated 24-well culture plates with 0.5 ml/well viral supernatant for 2 hr (3000 g, 4°C). The next day a second round of transduction was performed. After overnight incubation, the transduction rate was determined and a population of cells containing 1 x $10^6$ P14 TCR-expressing CD8 T cells was transferred into mice (C57BL/6) that were treated 1 d before with total body irradiation (TBI, 5 Gy, X-Ray irradiator RS2000, Rad Source). Ten days after T cell injection, the mice received 7.2 × 105 (*Johnson et al., 2009*) IU Proleukin (Novartis) intraperitoneally twice a day for 3 d and were monitored thoroughly (weight, appearance, behavior). Group assignments were blinded and mice were sacrificed according to predefined human endpoints.

## Flow cytometry assay for single chain mispairing

Human PBMCs were activated with anti-CD3/anti-CD28 and transduced with RD114-pseudotyped virus as described above. To determine whether epitope tags affect TCR surface expression, T cells transduced with F5 wtTCRα (± cMyc tag) and wtTCRβ (± V5 tag) were analyzed by flow cytometry using anti-cMyc and anti-V5 (Genscript, Piscataway, NJ) and anti-TCRβ (clone JOVI.1) primary antibodies, followed by anti-mouse IgG-PE secondary antibody. To measure mispairing of a single

introduced F5 TCR chain with endogenous human TCR chains, T cells transduced with cMyc-wtTCRα, cMyc-dsTCRα, V5-wtTCRβ, or V5-dsTCRβ individually were analyzed for surface expression of introduced chains with anti-tag antibodies.

Similar analysis was performed with mouse OTI TCR in mouse T cells, but tags were not required due to the availability of antibodies against the variable domains of OTI TCR (anti-mouse $V_\alpha 2$ (clone B20.1, Biolegend) and anti-$V_\beta 5.1/5.2$ (clone MR9-4, Biolegend). Fresh mouse splenocytes were depleted of T cells expressing endogenous TCRs with these variable domains by binding them with anti-$V_\alpha 2$-PE and anti-$V_\beta 5.1/5.2$ PE and then removing bound cells using anti-PE microbeads and MACS LD Separation Columns (Miltenyi Biotec, San Diego, CA) according to the manufacturer's instructions. Remaining T cells were activated with ConA/IL-7 and transduced as described above with ecotropic virus encoding LNGFR (transduction marker) and either OTI wtTCRα, dsTCRα, wtTCRβ, or dsTCRβ. Transduced T cells (gated on $CD45.1^+CD8a^+LNGFR^+$) were analyzed for surface expression of $V_\alpha 2$ and $V_\beta 5.1/5.2$ to measure mispairing of these single introduced OTI chains with endogenous murine TCR chains.

Analysis of mispairing of P14 TCR chains ($V_\alpha 2$ and $V_\beta 8$) ± codelivered shRNA was conducted identically except that 1) recipient cells were depleted of endogenous $V_\alpha 2^+$ and $V_\beta 8^+$ cells prior to transduction, and transduced cells were analyzed with anti-$V_\alpha 2$ and anti-$V_\beta 8$ (clone MR5-2, Biolegend) and 2) transduced cells were cultured in media with 10 ng/mL IL-15 for 72 hr following transduction and prior to flow cytometry analysis to enable endogenous gene knockdown by shRNA. Extent of endogenous TCR knockdown was determined by flow cytometry, comparing the amount of TCR complex (anti-TCRβ (clone H57-597) and anti-CD3ε (clone 145-2C11)) expressed on the surface of T cells transduced with LNGFR ± shRNA

### Statistics

Significance analysis of mouse survival (Kaplan-Meier) curves was conducted with the log-rank (Mantel-Cox) test. Significance analysis of mouse weight loss was conducted with one-way analysis of variance and Tukey's post-test for multiple comparisons. A statistical probability of $p < 0.05$ was considered significant. No power analysis was conducted for animal experiments; the number of animals tested was based on budget and feasibility. No data were excluded as outliers. Descriptive statistics are presented as mean ± standard deviation of replicate assays. Technical replicates are defined as replicate testing of tissue from the same mouse (only *Figure 4—figure supplement 3*) or identical assays conducted on independent manipulations of the same cell line or primary T cell sample (e.g. activity assays employ T cell replicates transduced separately and coincubated separately with target cells). Biological replicates are defined as identical experiments conducted in multiple mice.

### Acknowledgements

We thank Eugene Barsov and Richard Morgan (NCI) for providing the MSGV vector, Alok Joglekar (Caltech) for providing the pCCLc vector, and Gavin Bendle (Univ. of Birmingham) for providing the pMX vector. We also thank Gavin Bendle and Lili Yang (UCLA) for troubleshooting discussions related to mouse and human T cell transduction, Mireille Toebes (NKI) for helpful discussions related to tetramer production, and Owen N. Witte (UCLA) for use of his cell sorter. Short-dated Proleukin was provided without charge by Prometheus Laboratories Inc (San Diego, CA) through an investigator-initiated trial program. Biotinylated H-2K$^b$/Ova$_{257-264}$ monomers were provided by the NIH Tetramer Core Facility (Atlanta, GA). We gratefully acknowledge the staff of the Caltech animal facility for animal husbandry. This work was supported by the U.S.A. National Institutes of Health grant 5 P01CA132681-5 to DB and the Prostate Cancer Foundation Challenge Award 15CHAL02 to DB and MTB is the recipient of a Jane Coffin Childs Postdoctoral Fellowship. MSK is a Pew Scholar supported by the Pew Charitable Trusts.

# Additional information

## Competing interests

MTB, MHG and DB: Co-inventor on a patent application concerning the described technology (USPTO patent application 20150197771). The other author declares that no competing interests exist.

## Funding

| Funder | Grant reference number | Author |
| --- | --- | --- |
| National Institutes of Health | 5P01CA132681-5 | David Baltimore |
| Prostate Cancer Foundation | 15CHAL02 | Michael T Bethune<br>David Baltimore |
| Jane Coffin Childs Memorial Fund for Medical Research | | Michael T Bethune |
| Pew Charitable Trusts | | Michael S Kuhns |

The funders had no role in study design, data collection and interpretation, or the decision to submit the work for publication.

## Author contributions

MTB, MHG, MB, Conception and design, Acquisition of data, Analysis and interpretation of data, Drafting or revising the article; MSL, EHG, JZ, Acquisition of data, Analysis and interpretation of data, Drafting or revising the article; MSP, DC, Acquisition of data, Drafting or revising the article; JRH, DBK, Analysis and interpretation of data, Drafting or revising the article; MSK, WU, DB, Conception and design, Analysis and interpretation of data, Drafting or revising the article

## Author ORCIDs

David Baltimore, http://orcid.org/0000-0001-8723-8190

## Ethics

Animal experimentation: This study was performed in strict accordance with the recommendations in the Guide for the Care and Use of Laboratory Animals of the National Institutes of Health. Animal experiments conducted at Caltech were approved by the Institutional animal care and use committee (IACUC) of the California Institute of Technology (IACUC protocol 1611). Animal experiments conducted at UCLA adhered to UCLA Department of Laboratory Animal Medicine Standards (UCLA Animal Research Committee protocol #2008-167). Animal experiments conducted at MDC Berlin were conducted in accordance with institutional and national guidelines, and approved by a state animal ethics committee (Landesamt für Gesundheit und Soziales protocol 0233/11).

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
