## [Decision Letter]

Thank you for submitting your article "Domain-swapped T cell receptors improve the safety of TCR gene therapy" for consideration by *eLife*. Your article has been favorably evaluated by Sean Morrison as the Senior Editor and three reviewers: Brian Baker, Bernard Malissen, and a member of our Board of Reviewing Editors.

The reviewers have discussed the reviews with one another and the Reviewing Editor has drafted this decision to help you prepare a revised submission.

Summary:

This interesting manuscript builds and tests domain-swapped chains of the T cell receptor heterodimer to demonstrate that a number of domain-swapped combinations permit reasonable maintenance of assembly and function upon heterologous expression, yet substantially prevent assembly with 'wild-type' endogenous chains and the functional consequences thereof. The findings are of interest both for adding to the understanding of the structural constraints on T cell receptor assembly and signaling, and for creating a novel technological modality to add to the growing portfolio of methods for tumor-directed immunotherapy.

Essential revisions:

This is an exciting manuscript, but there are a few concerns we'd like you to address:

1) One concern relates to the somewhat sketchy quantification of the comparative efficiency of the domain-swapped T cell receptors. Specifically, there are two distinct ultimate functional outcomes in vivo being measured, and are correlatable with two distinct consequences of domain-swapped TCRs (dsTCRs); – reduction of pairing with endogenous TCR chains correlatable with loss of GVH induction activity, and reduction of cell-surface dsTCR levels potentially correlatable with possible quantitative reduction of anti-tumor activity. Quantified estimates of these correlations, perhaps most simply using cell number titrations for the relevant assays in vivo, would be immensely useful and very feasible additions.

2) A second concern is an extension of the first point above; – there are at least two other modalities for achieving these ends that the authors use for comparison; – stoichiometric expression of heterologous TCR chains for preferential coupling using a viral IRES substitute (2A), and shRNA-mediated 'knockdown' of endogenous TCR expression. Quantified estimates of the comparative efficiency of the dsTCR approach with these approaches for both GVH prevention and anti-tumor activity, again perhaps most simply using input cell number titrations for the relevant assays in vivo, would be very useful and feasible additions.

3) A major question in the manuscript is how the successful swapping strategies impact TCR/CD3 architecture on T cells, and what the authors perceive to be the 'loss of stabilizing contacts in the extracellular domains. It is unclear what this interpretation is based on. Figure 1 outlines the oft-discussed charges in the TM helices and how they are believed to interact and help align the CD3 chains, but it is not necessarily the case (albeit possible) that swapping these around would lead to a loss of stabilizing contacts in the extracellular parts, given what we know (and don't know) about the structural relationships and the constraints on TCR/CD3 overall architecture. In fact, it is plausible to argue that the data suggest that there is malleability in the architecture (and by extension, some malleability in the signaling mechanism), consistent with some published literature. A related issue to address is the swapping of the actual constant domains themselves – this is likely to have considerable disruptions simply based on structural principles of how the constant domains interface with each other and the variable domains. It would be useful to discuss these issues to have them better clarified in the manuscript.

---

## [Author Response]

*[…] Essential revisions:*

*This is an exciting manuscript, but there are a few concerns we'd like you to address:*

*1) One concern relates to the somewhat sketchy quantification of the comparative efficiency of the domain-swapped T cell receptors. Specifically, there are two distinct ultimate functional outcomes in vivo being measured, and are correlatable with two distinct consequences of domain-swapped TCRs (dsTCRs); reduction of pairing with endogenous TCR chains correlatable with loss of GVH induction activity, and reduction of cell-surface dsTCR levels potentially correlatable with possible quantitative reduction of anti-tumor activity. Quantified estimates of these correlations, perhaps most simply using cell number titrations for the relevant assays in vivo, would be immensely useful and very feasible additions.*

*2) A second concern is an extension of the first point above; there are at least two other modalities for achieving these ends that the authors use for comparison; stoichiometric expression of heterologous TCR chains for preferential coupling using a viral IRES substitute (2A), and shRNA-mediated 'knockdown' of endogenous TCR expression. Quantified estimates of the comparative efficiency of the dsTCR approach with these approaches for both GVH prevention and anti-tumor activity, again perhaps most simply using input cell number titrations for the relevant assays in vivo, would be very useful and feasible additions.*

Thank you for this suggested experiment. In response to these two points, we have performed a follow-up study in which we transduced mouse splenocytes with ovalbumin-specific OTI wtTCR, dsTCR, or codon optimized wtTCR with shRNA-mediated knockdown of endogenous TCR. In all of these constructs, the TCR α and β chain genes were linked using a 2A sequence. We then injected mice with varying numbers (3.3x10^5^, 10^5^, or 10^4^) of transduced T cells (down from 10^6^ cells used in manuscript). Two experiments were then performed:

A) Animals were first tested for GvHD as described in the manuscript. At these lower input levels, none of the animals developed disease. This was expected because the 2A linker ensured equal stoichiometry between the introduced α and β chains – decreasing mispairing (Figure 4 and previously reported^1^) – and because lower polyclonality of the transduced T cell population reduces the risk of deleterious mispairing^1,2^.

B) Next, we injected animals with 5x10^6^ ovalbumin-expressing E.G7 thymoma and monitored the growth of these tumor cells over time. At 10^5^ T cell input and higher, wtTCR- and dsTCR-transduced T cells similarly prevented tumor growth in all mice tested. At 10^4^ T cell input, both wtTCR and dsTCR-transduced T cells slowed but did not prevent tumor growth entirely. This suggests that wtTCR- and dsTCR-transduced T cells are similarly efficient at protecting against tumor growth in this xenograft model. These results have been added to the manuscript as Figure 4—figure supplement 4, and have been discussed in the revised text.

*3) A major question in the manuscript is how the successful swapping strategies impact TCR/CD3 architecture on T cells, and what the authors perceive to be the 'loss of stabilizing contacts in the extracellular domains. It is unclear what this interpretation is based on. Figure 1 outlines the oft-discussed charges in the TM helices and how they are believed to interact and help align the CD3 chains, but it is not necessarily the case (albeit possible) that swapping these around would lead to a loss of stabilizing contacts in the extracellular parts, given what we know (and don't know) about the structural relationships and the constraints on TCR/CD3 overall architecture. In fact, it is plausible to argue that the data suggest that there is malleability in the architecture (and by extension, some malleability in the signaling mechanism), consistent with some published literature. A related issue to address is the swapping of the actual constant domains themselves – this is likely to have considerable disruptions simply based on structural principles of how the constant domains interface with each other and the variable domains. It would be useful to discuss these issues to have them better clarified in the manuscript.*

There are two questions asked in point 3 above:

First, how do we know extracellular contacts are disrupted by domain-swapping?

We do not know. We hypothesize that this is so based on observations that 1) transmembrane contacts are critical for assembly^3^ so these must be preserved in the dsTCR/CD3 complex, 2) extracellular variable (V) and constant (C) domains are rotated relative to TM domains in dsTCR constructs, so if CD3 contacts with TCR transmembrane domains are preserved, it is unlikely that the short, rigid connecting CD3 peptides enable the extracellular contacts to be preserved in the reoriented TCR C domains, and 3) dsTCR/CD3 complex assembly/surface expression is impaired relative to wtTCR/CD3 complex assembly, so disruption of extracellular contacts is a potential (unproven) hypothesis to explain this impairment.

Second, how does domain-swapping affect the structural relationship between variable and constant domains?

Figure 2 and Figure 7 suggest that the relationship between the V and C domains sets constraints on domain-swapping. For both constructs in which we attempted to position the domain-swap directly between the extracellular V and C domains, we ablated surface expression of the dsTCR complex entirely. It seems likely that the conserved, rigid hinges of the TCR chains do not enable proper V-V and C-C pairing in the domain-swapped constructs. Only by domain-swapping further down the chains – at junctions before and after the connecting peptide – were we able to get functional dsTCR complexes.

In response to this comment, we have expanded and clarified our discussion of TCR/CD3 assembly in the wt and ds complexes. We have also emphasized that our presumption of ectodomain-CD3 contact disruption is a hypothesis only, and is undemonstrated in the current work.

1) Bendle, G.M., et al. Lethal graft-versus-host disease in mouse models of T cell receptor gene therapy. Nat Med 16, 565-570, 561p following 570 (2010).

2) Heemskerk, M.H., et al. Reprogramming of virus-specific T cells into leukemia-reactive T cells using T cell receptor gene transfer. The Journal of experimental medicine 199, 885-894 (2004).

3) Call, M.E., Pyrdol, J., Wiedmann, M. & Wucherpfennig, K.W. The organizing principle in the formation of the T cell receptor-CD3 complex. Cell 111, 967-979 (2002).